# Proximity-Based Optical Camera Communication with Multiple Transmitters Using Deep Learning

**DOI:** 10.3390/s24020702

**Published:** 2024-01-22

**Authors:** Muhammad Rangga Aziz Nasution, Herfandi Herfandi, Ones Sanjerico Sitanggang, Huy Nguyen, Yeong Min Jang

**Affiliations:** Department of Electronics Engineering, Kookmin University, Seoul 02707, Republic of Korea; rangganast@gmail.com (M.R.A.N.); herfandi@uts.ac.id (H.H.); onessanjerico@gmail.com (O.S.S.); ngochuy@kookmin.ac.kr (H.N.)

**Keywords:** optical camera communication, proximity, multiple transmitters, object detection

## Abstract

In recent years, optical camera communication (OCC) has garnered attention as a research focus. OCC uses optical light to transmit data by scattering the light in various directions. Although this can be advantageous with multiple transmitter scenarios, there are situations in which only a single transmitter is permitted to communicate. Therefore, this method is proposed to fulfill the latter requirement using 2D object size to calculate the proximity of the objects through an AI object detection model. This approach enables prioritization among transmitters based on the transmitter proximity to the receiver for communication, facilitating alternating communication with multiple transmitters. The image processing employed when receiving the signals from transmitters enables communication to be performed without the need to modify the camera parameters. During the implementation, the distance between the transmitter and receiver varied between 1.0 and 5.0 m, and the system demonstrated a maximum data rate of 3.945 kbps with a minimum BER of $4.2\times 10^{-3}$. Additionally, the system achieved high accuracy from the refined YOLOv8 detection algorithm, reaching 0.98 mAP at a 0.50 IoU.

## 1. Introduction

Wireless communication has found widespread use across various fields and has been seamlessly integrated into daily life over the past few decades. In contrast to wired communication, which relies on cables and can be inconvenient and cumbersome, wireless communication provides flexibility and convenience by eliminating the need for cables. Wireless systems predominantly rely on radio frequency (RF) communication networks [1]. Presently, the research focus has shifted toward sixth generation (6G) cellular networks, which exhibit the potential to deliver data rates ranging from 1 to 10 Tbps [2]. Despite the growing interest in 6G, it is imperative to acknowledge the numerous disadvantages associated with RF-based communications. Human bodies are prone to electromagnetic waves emanating from RF systems [3]. Furthermore, the emergence of wireless communication technology may contribute to the depletion of RF waves during the formation process [4]. Conversely, the past decade has witnessed a surge in the usage of mobile devices such as smartphones, tablets, and sensors, intensifying the demand for a higher bandwidth spectrum [5]. Optical wireless communication (OWC) presents a potential solution to this problem, leveraging the vast and unregulated bandwidth of optical light, spanning 200 THz in the 700–1500 nm range [6].

Optical camera communication (OCC) is a sub-topic of OWC, where it uses light source devices such as light-emitting diodes (LEDs) or display screens as transmitters and cameras as receivers [7]. OCC involves encoding data in visual signals emitted by a light source and decoding it through a camera for inter-device communication [8]. Notably, OCC presents several advantages over RF-based communication [9]. The use of visible light in the OCC scheme is considered harmless to human health [10]. In contrast to RF, OCC uses visible light as the communication medium, offering enhanced robustness against interference and jamming, thereby ensuring heightened security in communication [11]. Furthermore, OCC provides flexibility with lower cost, lower complexity, and lower power consumption [12]. The merits of OCC have prompted the IEEE’s interest, resulting in the formation of the IEEE 802.15.7m task group dedicated to further exploration in this field [13].

OCC technology employs several devices as transmitters in the communication system with a light-emitting diode (LED) being a notable example. Apart from its role in OCC, LEDs are also used in visible light communication (VLC) technology [14]. In VLC, unlike OCC, a photodiode serves as the receiver, functioning as a semiconductor device capable of detecting light waves and converting them into electrical currents. In the VLC system, the electrical current generated by the photodiode serves as the communication signal [15]. In contrast, the OCC system uses visible light as the communication signal [16]. To enhance the data capacity on the transmitter side, multiple LEDs are employed, forming LED matrices with various sizes, such as 8 × 8, and 16 × 16. The use of an LED matrix, which is capable of accommodating more signals, offers distinct advantages over a single LED configuration [17,18].

Similar to RF-based communication, OCC operates in a broadcast manner albeit within a localized environment. This implies that the OCC can transmit data to multiple receivers and vice versa. Consequently, in the OCC scheme, employing multiple transmitter devices as different entities to transmit the data to the receiver is feasible. While this is advantageous for scenarios requiring multiple simultaneous communication, it may be disadvantageous in schemes where only a single transmitter is permitted, such as for security purposes. Consequently, specific techniques are required to enable OCC to facilitate one-to-one directional communication, even in the presence of multiple active transmitters.

Distance serves as a parameter to determine which transmitter is authorized to send data to the receiver in OCC. Measurement of the object’s z-plane position from the camera facilitates distance determination and aids in establishing communication priority. However, measuring distance in a 2D image poses challenges, because there is no z-plane in the 2D dimension, which is inherent in a 3D environment. Research conducted by John and Laurence [19] indicates that, given an object in a specific position within a 2D image, human eyes may struggle to accurately infer the distance of the object, even when provided with the real-world size of the object. However, human eyes and perception can effectively discern the proximity of the objects in the visual field. As depicted in Figure 1, In the real world, objects appearing smaller typically indicate a greater distance compared with objects with larger dimensions. Cameras, while not equivalent to human eyes, share similar fundamental functions. Both aim to capture the 3D view of a scene and subsequently convert it into a 2D representation.

In this study, we propose a 2D proximity-based method to determine the appropriate LED for communication. Through this method, secure data transmission in the OCC system can be achieved. Despite the simultaneous presence of multiple transmitters, the data will be sourced exclusively from the selected and determined transmitter. Furthermore, this method allows for interchangeable transmission systems. While one transmitter is actively communicating, another transmitter can be activated, leading to the deactivation of the previously active transmitter.

The overall contributions of this study are as follows:Introducing a method prioritizing LED transmitters in scenarios with multiple transmitters and a single receiver based on the proximity of the transmitters to the receiver.Proposing a method for an interchangeable transmission system with multiple transmitters, using 2D object size for measuring proximity of the transmitters.Introducing an approach to read the LED array data without the need for camera parameter modification.

This paper is organized as follows. Section 2 highlights recent studies on multiple-transmitter single-receiver OCC schemes, and Section 3 describes the proposed method. Section 4 elaborates on the experiment and results analysis. Section 5 provides the conclusion of the study.

## 2. Literature Review

Numerous studies have explored multiple-transmitter single-receiver schemes. Jing et al. [20] introduced a method for the multi-LED scenario in a mobile OCC system (MOOC). Their work aimed to mitigate the overlapping issues that arise when OCC is implemented in mobile environments. In instances where a receiver traverses intersections of multiple LEDs, the received frames may overlap, resulting in irregular received data. The authors proposed a multi-column matrices selection (MCMS) method that integrates k-means clustering algorithms. Their approach addresses signal interference and mitigates performance degradation resulting from complementary metal-oxide-semiconductor (CMOS) camera movement. This research yielded a notable enhancement in BER performance. At a moving speed of 20 cm/s with a distance of 20 cm between two LEDs, the BER was measured at $8.74\times 10^{-6}$.

Arai et al [21] proposed a method for determining the position of multiple transmitters in an infrastructure-to-vehicle visible light communication I2V-VLC system. In this approach, various traffic-related infrastructures, such as traffic lights and brake lights in cars, were used as transmitters. The receiver was a camera affixed to the front part of a car, which captured information from the aforementioned transmitters. Image clipping was employed to capture the LED arrays in a frame with each LED array processed individually. This method uses a block-matching algorithm to determine the position of an LED array, which is calculated from one frame to another in sequence. The proposed method achieved an almost perfect success rate in detecting multiple LED arrays through FN and FP calculations.

Ifthekhar et al. [22] conducted research on cooperative vehicle positioning using OCC. Commonly, the position of vehicles is determined through global positioning system (GPS). However, the usage of GPS may be distracted when the vehicles enter the tunnels. OCC is utilized due to its resistance to jamming and disruption, especially in area that GPS cannot reach into. The proposed method uses multiple LEDs and multiple cameras to communicate each other. In this study, the position of the vehicle is estimated using two methods: the neural-network-based method and computer-vision method. The result shows that the neural-network-based approach estimates the position of the vehicle better than the computer-vision based method, assumed the terrain is flat and the height of the vehicles are level.

The aforementioned research substantiates the possibility of implementing OCC in a multiple-transmitter single-receiver scheme. The use of several transmitters offers advantages such as increased data rates and enhanced transmission capability. Despite existing studies on related topics, the investigation of priority-based multiple transmitters remains unexplored. In OCC, the camera’s ability as a receiver to capture signals from several transmitters is possible, underscoring the importance of maximizing its potential. However, in several scenarios, communication may not involve all transmitters simultaneously. Therefore, this study proposes a method to measure priority given the scenarios mentioned above. By calculating the object size in 2D representation and leveraging an object detection AI model, it becomes feasible to determine which transmitter is authorized to transmit data to the receiver. The proposed method also enables alternate communication.

## 3. Description of the Proposed Method

This section describes the method to measure communication priority using the transmitter proximity in OCC. Recent research has extensively explored multiple-transmitter single-receiver schemes in OCC systems. Figure 2 illustrates the working principle of this method. As previously mentioned, there are scenarios in which multiple transmitters are employed to transmit different signals from different entities. This method is specifically designed for scenarios that restrict the receiver from receiving signals except from a designated transmitter even in the presence of several transmitters in coverage.

The method operates by continuously measuring the transmitters’ proximity to the receiver. Proximity was measured using bounding boxes derived from the YOLO object detection model. In this context, the YOLO object detection method is employed on the receiver side to identify and visualize the location of objects or transmitters within an environment. These bounding boxes serve as a real-time visual representation of the objects’ locations in the image, and using information from this detection model, the method can identify objects on the transmitter side to accurately measure their proximity to the receiver. Based on the previously explained principle of proximity, we designed a method for the multiple-transmitters single-receiver OCC scenario, using the bounding boxes generated by the YOLO model as previously mentioned. Given that OCC uses a camera as the receiver, the aforementioned principle can be applied to determine the relative distances between transmitters. Using the bounding box, the size of the LED array transmitters can be measured, and the difference in transmitter sizes can be determined. In this study, the transmitter deemed “closer” to the receiver is the one authorized to transmit data to the receiver.

### 3.1. Transmitter

In this method, we use multiple LED array transmitters to simulate an advanced communication system. Each LED array serves as a channel for transmitting a digital signal to the receiver camera. The data are encoded into binary numbers “0 s” and “1 s”, which is a prevalent practice in digital signal processing (DSP) that is easily interpreted by computers. Employing on–off keying (OOK) modulation, where “1 s” and “0 s” are represented by the states of “on” and “off” on the LED, enables efficient data communication to the receiver. In OCC systems where LED arrays function as transmitters, the abundance of LEDs enables each to operate as a dedicated channel, independently transmitting a single bit of data. This approach optimizes the overall efficiency and data throughput of the system, highlighting its potential for robust and high-capacity communication.

The subsequent phase involves the incorporation of the sequence number (SN), which is a crucial component in a communication system that distinguishes received data packets at the receiver. The inclusion of SN also enables the receiving cameras to detect any missing packets, particularly in cases of oversampling. It designates the payload, encapsulating sequence information for each data packet, with the flexibility to adjust the length of the SN according to the system conditions. Although channel conditions may influence the SN length, it can be truncated to enhance system performance.

In the proposed method, multiple LED transmitters are used in the OCC scheme. Employing sequence numbers, each transmitter receives a unique identifier to distinguish it from the others. Positioned in front of the camera, the transmitters change positions forward and backward, simulating the distance difference between LEDs. The transmitter closer to the camera will exhibit a larger size, signifying its authorization to transmit data to the receiver. The LED array transmitter mapping in this proposed method is illustrated in Figure 3.

### 3.2. Receiver

On the receiver side, the camera captures multiple LED array transmitters. Achieving accurate tracking of each LED matrix requires the combined use of the YOLOv8 object detection model and OpenCV tracker. The YOLOv8 object detection model ensures reliable initialization when detecting objects. Consequently, in this scheme, the YOLOv8 model is employed to generate regions of interest (ROIs) on the LED matrix transmitters. Given that the YOLOv8 object detection model consumes considerable resources and exhibits relatively intensive processing demands, especially with moving objects, the OpenCV tracker is employed to monitor the movement of the LED matrix transmitters. OpenCV offers several lightweight trackers capable of tracing position changes in transmitters. However, a disadvantage of using this tracker is that the tracker size may fluctuate during continuous tracking. To mitigate this issue, both the YOLOv8 model and the OpenCV tracker are used simultaneously. This dual approach ensures that the ROI maintains its shape and does not undergo undesired enlargements or reductions throughout the tracking process. The overall detection approach is depicted in Figure 4.

Next, each frame is processed on the receiver side using the OpenCV library. In contrast to previous research [1,11] that modified the camera parameters approach to perform OCC, this study does not employ the camera parameters modification approach. Instead, each received frame is processed using OpenCV. This choice is attributed to the use of the YOLOv8 object detection model, which is fine-tuned with images featuring LED matrices to establish the ROI. Therefore, modifying camera parameters will impede the object detection model’s ability to detect the presence of LED transmitters.

Rather than adjusting the camera parameters, modifications are applied when capturing the frame of the detected LED. First, a scaler transformation is used to convert pixels into absolute values, which is followed by the conversion of pixels into unsigned 8-bit integers. Next, a Gaussian blur [23] is applied, leveraging the Gaussian function to blur the image. This blurring technique is used to reduce the image noise and details. Given that LEDs emit light in a scattering manner, a typical amount of light noise is produced. Therefore, Gaussian blur facilitates the removal of unnecessary details and noise, retaining only the essential pixels. The Gaussian blur function is formulated in Equation (Equation 1). Subsequently, grayscale transformation is executed on the scaled frame to facilitate processing and mitigate the scattering light noise effect produced by the LEDs.
(1)$$G\left(x,y\right)=\frac{1}{2\pi \sigma^{2}}e^{-\frac{x^{2}+y^{2}}{2\sigma^{2}}}$$

After grayscale transformation, a binary thresholding method is employed to convert the frame to a binary representation. Pixels with intensities exceeding a certain threshold are assigned one value (typically 255 for white), while pixels below the threshold are assigned another value (usually 0 for black). This thresholding process yields several shapes. The contours of the shapes are derived and bounded within rectangular shapes. Only rectangular shapes that fall within predetermined sizes are considered. The final step involves the decoding process. Decoding the LEDs entails determining the position of the LEDs in one row by measuring the distances between the rectangular shapes. The distances between the LEDs can be determined by calculating the width of the rectangular shape and the distance between one shape and another. The overall frame processing is depicted in Figure 5.

## 4. YOLOv8 Object Detection

Object detection, within the realm of computer vision, is a task that focuses on accurately identifying the locations of objects in a 2D representation. This intricate process involves categorizing objects into specific classes, ranging from vehicles and aircraft to animals and individuals, thereby facilitating a comprehensive understanding of visual scenes [24].

YOLO employs a one-stage detection approach, indicating that both bounding box localization and object identification occur directly within a single feed-forward fully convolutional network [25]. In contrast, two-stage detection [26] divides the detection process into two distinct phases. In the initial phase, object candidates are proposed through the ROI, while the second phase focuses on identifying objects within the proposed ROI candidates. Although the two-stage approach is often considered for its ability to produce more precise detections, it is comparatively slower because of the preprocessing involved in the initial stage [27]. Consequently, the one-stage method is deemed more practical for mobile scenarios because of its lightweight and convenient results.

The YOLO model comprises a backbone, a neck, and a head. The backbone is a pre-trained convolutional neural network designed to extract image features such as ResNet [28] or DarkNet [29]. The neck acts as a connector between the backbone and head. Commonly used pre-trained neural networks in YOLO include the feature pyramid network (FPN) [30] and the path aggregation network (PAN). The head is the part where detection is performed. Detection involves splitting the image into S × S grids and predicting bounding boxes (B) and class probabilities (C) for each grid [31]. The class-specific confidence score is expressed by Equation (Equation 2).
(2)$$\mathrm{pred}={\mathrm{Pr}(\mathrm{Class}}_{i})\times {\mathrm{IOU}}_{\mathrm{truth}}$$

YOLOv8 introduced a novel detection model approach by implementing anchor-free detection and enhancement of mosaic data [32,33]. The anchor-free detection method resulted in a reduced number of squares to be predicted, thus expediting the process of non-maximum suppression (NMS) and enhancing the mean average precision (mAP) of the detection process. This result is obtained by substituting the C2f module with the C3 module. YOLOv8 also introduced a detection model approach to reduce training time and swiftly identify multiple objects. This is achieved by employing the technique of consolidating multiple objects in a figure into one, which is used as the model’s input [11].

As shown in Figure 6, YOLOv8 consists of multiple layers of architecture [34], namely backbone, neck, and head (prediction). The backbone is the layer responsible for preprocessing image data and is composed of several convolutional neural networks. In YOLOv8, CSPNet [35] serves as the backbone. In the subsequent stages of YOLOv8, the neck includes concats and upsample layers alongside regular layers such as C2f and convolutional layers. In the head, three detection modules are used, and the output is assembled based on the three detection modules. The overall YOLOv8 architecture is illustrated in Figure 6.

YOLOv8 offers five different variants: YOLOv8n, YOLOv8s, YOLOv8m, YOLOv8l, and YOLOv8x. The differences between these variants lie in the number of hidden layers and the complexity of their backbone networks. Larger models require more computational resources for object detection. In this study, the YOLOv8 model is also compared to other object detection methods including the YOLOv5 model and Faster-RCNN [36], which is the best object detection that adopts the two-stage detection approach.

## 5. Experimental Setup

The experiment involved employing two LED arrays positioned in parallel along the x-axis in front of the camera. Both LED arrays were of identical size, and each LED array comprised 64 LEDs on the board, which were arranged in an 8 × 8 matrix. In this setup, as illustrated in Figure 3, the LED sequence number and identifier were assigned to the first and last rows, while the second row was used to transmit text data. OOK modulation was used to map the data on the transmitter row. On the receiver side, a camera was used to capture both LED arrays. All hardwares used in this experiment are depicted in Figure 7.

As previously mentioned, in this experiment, the camera’s parameters remained unmodified because of the use of the YOLOv8 model. The visibility of the LED array was essential for the object detection model to create an ROI. The experiment used Python and C++ programing languages, along with various deep learning frameworks such as Torch and ONNXRuntime, as the inferencing library. The hardware and software details for the simulation and training are provided in Table 1.

In simulating interchangeability based on 2D relative distance, a crucial consideration involved manipulating the sizes of LEDs positioned in front of a camera. This manipulation creates a distinction between LED arrays by varying their sizes and positioning them at different distances. The primary objective was to enable a specific LED array to transmit data to the receiver. This was achieved by strategically positioning the LED array further forward. Using a fine-tuned YOLOv8 object detection model, the process involved identifying and delineating the boundaries of both LED arrays. These boundaries were represented by bounding boxes, and the subsequent step involved comparing them by calculating the area size of each bounding box.

After successfully detecting both LED arrays, the system proceeded to evaluate the area sizes of their respective bounding boxes. This comparison was crucial for determining which LED array engaged in communication with the receiver. A bounding box with a larger area was selected as the target for data transmission. This approach ensures a dynamic and adaptable system that intelligently regulates the interchangeability of LED arrays based on their relative sizes and distances from the camera. The overall process, illustrated in Figure 8, represents a sophisticated approach to optimizing data communication in scenarios with multiple LED arrays.

As previously mentioned, this study employed YOLOv8 object detection to locate the LED array transmitters. In addition, the model was also used in the proposed method as a medium to measure LED array transmitter sizes in 2D images. Several YOLO variant models were fine-tuned using several images containing the LED array. The images used were generalized with the LED array positioned at various angles and distances from the camera. Furthermore, the dataset included images that featured either a single LED array or two LED arrays simultaneously. The dataset samples are illustrated in Figure 9.

In total, 750 image data were obtained with a total size of 2.5 GB. The data were split into training and validation sets with a ratio of 70% for training and 30% for validation. The dataset was randomly shuffled during the splitting process. The data were labeled using the CVAT labeling tool [37] employing bounding box shapes. After drawing bounding boxes on the images, the box properties (x and y coordinates, width, and height) were derived and normalized on the image ratio for YOLO fine tuning. Normalization on bounding box data is not applied on other object detection methods.

The pre-trained model was downloaded from the Ultralytics repository, and training was performed using the PyTorch library. Following fine-tuning of the pre-trained model, the PyTorch output format (.pt) was converted to the ONNXRuntime format (.onnx). This conversion enabled inferencing using the ONNXRuntime library and visualized more lightweight using the OpenCV library. The training required approximately 45 min on a system equipped with 16 GB RAM and an Nvidia RTX 3060Ti GPU.

## 6. Experimental Result and Discussion

### 6.1. Correlation between Proximity and Object Size for Priority Decision Making

Figure 10 illustrates the two LED array transmitters on the left and right sides. When the left LED is brought forward, a green box tracker appears, indicating that the left-hand side LED array is communicating with the camera. Similarly, moving the right LED array results in a blue rectangle tracker, signaling that the right-hand side transmitter is conveying information to the camera. This implies that the object size is correlated with the proximity in front of the camera. Figure 8 further substantiates this observation, as the distances from the front of the two transmitters are not identical.

This indicates that in 2D representation, even without knowing the exact distance of each transmitter from the receiver, priority in OCC can be determined based on the proximity of the transmitters. Furthermore, through 2D object size estimation, interchangeable communication can be performed. LED transmitters can also alternately communicate with a single receiver. The transmitter intending to communicate with the receiver can move forward and then backward when it decides to cease communication. This approach can be applied similarly to other transmitters.

Because no additional devices are employed, using an object detection AI model to establish priority in scenarios with multiple LED transmitters proves to be economical and less complex. This approach eliminates the need for sensor fusion applications. Furthermore, information from the object detection model is only used when the tracker fluctuates, such as when it fails to accurately trace the LED transmitter. Fluctuations are determined based on the boundaries of transmitters present in the first and last rows. Therefore, the object detection model is not used continuously, leading to minimal resource usage.

As depicted in Figure 10, text data are transmitted through the second row of the LED array. Despite potential disruptions in communication due to noise, data transmission can still be achieved, and the camera remains capable of detecting the OOK-modulated LEDs. Successful data reading is demonstrated by the presence of small blue squares that detect LEDs, indicating the data reading of each LED. Subsequently, these small squares are processed, and the proximity of the transmitters is calculated using the ratio of the bounding box tracker size.

However, the proposed method remains sensitive to changes in brightness, saturation, or contrast of the visual environment. As illustrated in Figure 11, in instances where the background’s brightness is low, the light emitted from the turned-on LEDs can affect the turned-off LEDs, leading to occasional misestimation of the turned-off LEDs as turned-on. Despite its sensitivity to brightness noise, this approach demonstrates the proper detection of turned-on LEDs under average brightness conditions.

### 6.2. Object Detection Performance

In this proposed method, the performance of the object detection model is crucial given its role in determining the LED transmitter size and establishing communication priority. Therefore, enhancing the performance of the object AI model is a critical step in the process.
(3)$$\mathrm{IoU}=\frac{B\cap B_{gt}}{B\cup B_{gt}}$$
(4)$$\mathrm{mAP}=\frac{1}{n}\sum_{k=1}^{k+n}AP_{k}$$
(5)$$\mathrm{Precision}=\frac{\mathrm{TP}}{\mathrm{FP}\;+\;\mathrm{TP}}$$
(6)$$\mathrm{Recall}=\frac{\mathrm{TP}}{\mathrm{FN}\;+\;\mathrm{TP}}$$

As detailed in Table 2, the training achieved satisfactory results with the YOLOv8 trained model achieving 0.98 mAP at 0.50 IoU. Compared to the previous YOLO, the accuracy did not reduce heavily and only fluctuated around 0.97 and 0.98 of mAP at 0.50 IoU. Moreover, compared to the two-stage algorithm, Faster-RCNN, the mAP does not have a large difference. The result indicates that using the one-stage algorithm is more advantageous. As mentioned before, the two-stage approach is slower than the one-stage approach due to the pre-processing stage. The usage of YOLOv8 here was more beneficial than using other algorithms. The training result also demonstrates the model’s successful and accurate detection of LED array locations in most of the validation data.

The model implementation demonstrated commendable performance in detecting the LED arrays in both single and multiple LED scenarios. As shown in Figure 10, when applied in the proposed scheme, the model accurately provides box properties for use in the tracker. In practice, the fine-tuned object-detection model enhanced the accuracy tracker, enabling precise and continuous communication.

### 6.3. Data Rate and BER Estimation

In this study, we conducted comparative experiments to evaluate the system’s performance. The performance of data transmission without modifying camera parameters was assessed using BER. The transmitted data are susceptible to imperfections caused by disruptions such as noise, distortion, or receiver anomalies. BER is determined by calculating the occurrence of false bits over a given period. After calculating the errors, the BER is obtained by dividing the number of bit errors by the total number of bits received.

Figure 10 displays the successful OCC experiment results based on our proposed method. Our system is capable of achieving data transmission speeds of up to 3.945 kbps with an 8 × 8 LED array configuration and can achieve higher data rates of up to 15.45 kbps with a 16 × 16 LED array configuration. However, this paper focuses on employing an 8 × 8 LED array.

The BER calculation in this study is conducted by estimating the level of character correction received and transmitted from the LED array. If the received character is incorrect, it is counted as a false positive for 8 bits. Table 3 presents the BER performance results of various methods proposed in other literature. It can be observed that the proposed model maintains a constant BER value at distances from 1 to 2.5 m unlike other methods whose BER performance is unstable at dynamic distances. Although at a distance of 5 m, there is a significant increase in BER for the proposed model, the obtained BER value still outperforms existing methods. Furthermore, the increase in BER is also influenced by the size of the LED array, as the farther the LED array detected by the camera, each LED array becomes smaller and harder to detect by the camera.

## 7. Conclusions

This paper proposes a method for prioritizing transmitters based on their size in 2D representation. When two objects exhibit similar 2D dimensions, their proximity can be deduced through object size measurement in both 2D representation and real-world scenarios. The correlation between 2D object size and object position arises from the tendency of closer objects to appear larger and farther to objects appear smaller. This method is suitable for scenarios requiring single transmitter communication with the receiver, especially when multiple transmitters are present in a single camera frame.

The proposed method uses object detection AI models, which excel in locating object positions by drawing bounding boxes around them. Given that an LED array serves as an OCC transmitter, its size can be accurately measured through object detection, leveraging the distinctive shape of the LED array. Therefore, in this study, the proximity of the LED arrays can be measured and priority can be established. This approach not only enables interchangeable data transmission but also serves as an economical technique for determining priority between transmitters without the need for additional devices.

The object detection model relies on the visible detection of the transmitters, preventing arbitrary modifications to the camera parameters. Consequently, a method needs to be developed to enable OCC without altering the camera parameters. The proposed method, which involves image frame processing in multiple steps, yields notable results with negligible errors, particularly within an average distance range of 1 to 5 m.

Future research should focus on extending the application of this method to simulate scenarios involving more than two transmitters simultaneously. Maximizing the camera’s potential by using multiple transmitters concurrently could be explored. In addition, further investigation into methods that enable higher data rate transmission without modifying camera parameters would be beneficial.

## Figures and Tables

**Figure 1 sensors-24-00702-f001:**
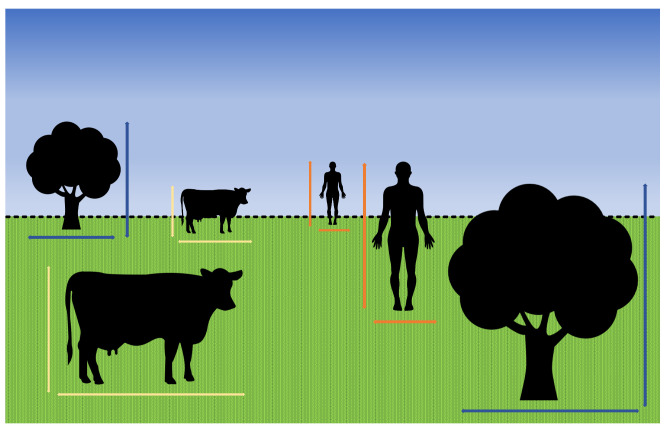
In 2D representation, several same-sized objects may have different 2D sizes if the objects are placed at different distances.

**Figure 2 sensors-24-00702-f002:**
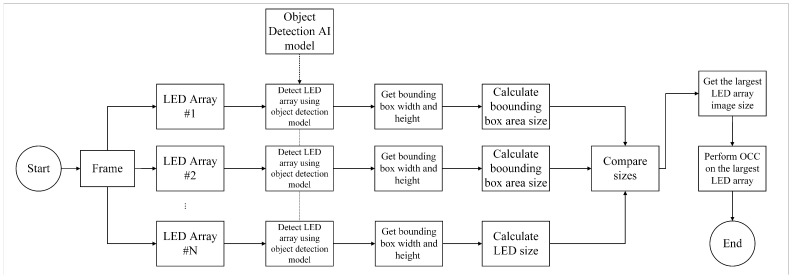
Multiple transmitter priority decision based on 2D object size.

**Figure 3 sensors-24-00702-f003:**
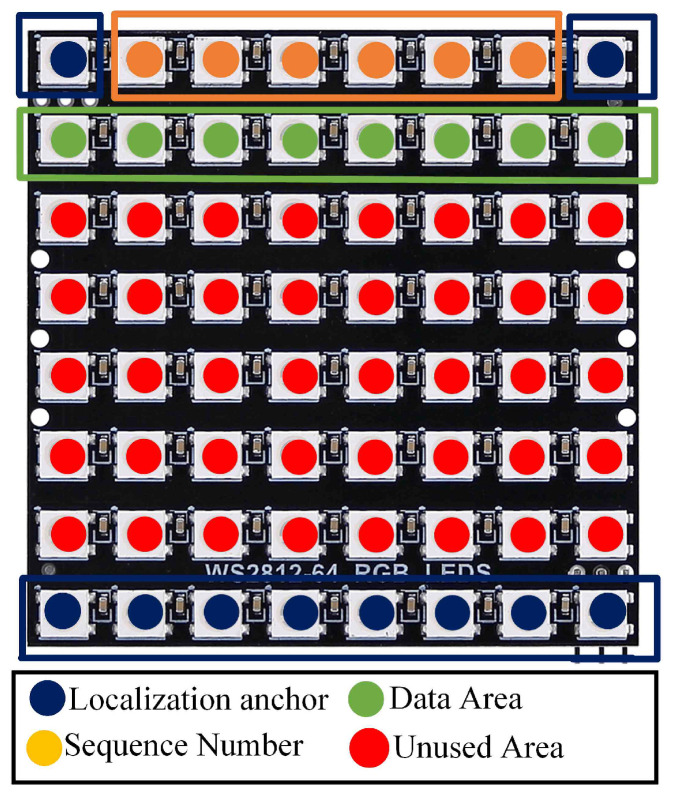
Mapping of LED array transmitter.

**Figure 4 sensors-24-00702-f004:**
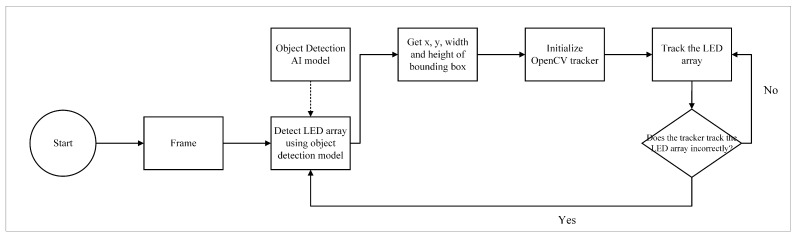
Hybrid OpenCV tracker and YOLOv8 approach for LED array detection.

**Figure 5 sensors-24-00702-f005:**
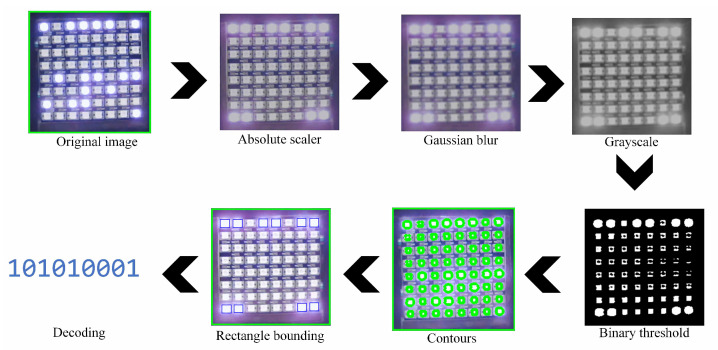
LED array frame processing transformation steps for the unmodified camera parameters scenario.

**Figure 6 sensors-24-00702-f006:**
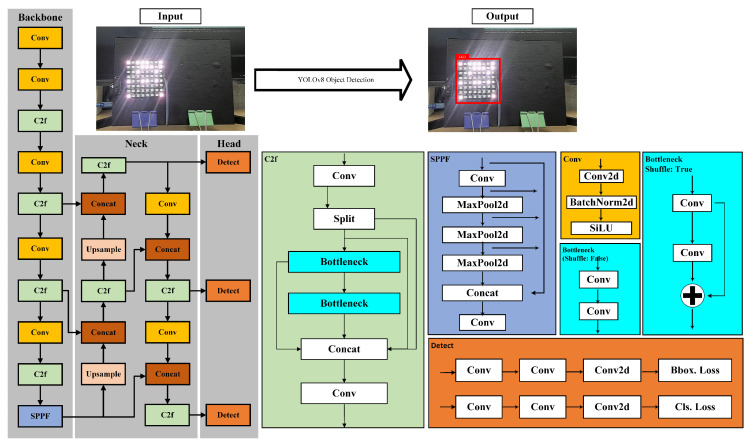
YOLOv8 object detection architecture.

**Figure 7 sensors-24-00702-f007:**
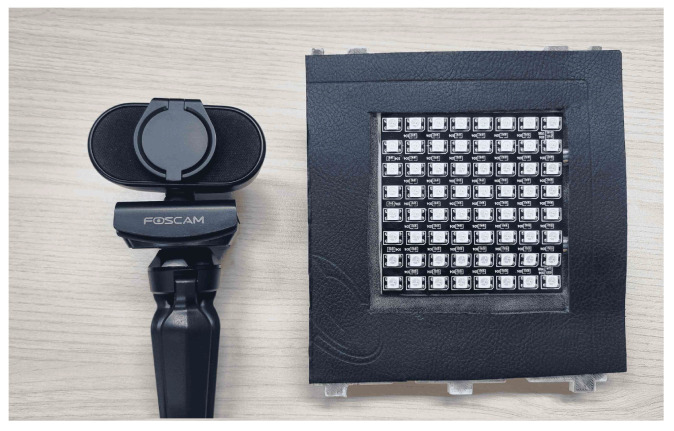
Hardware equipments used for simulation.

**Figure 8 sensors-24-00702-f008:**
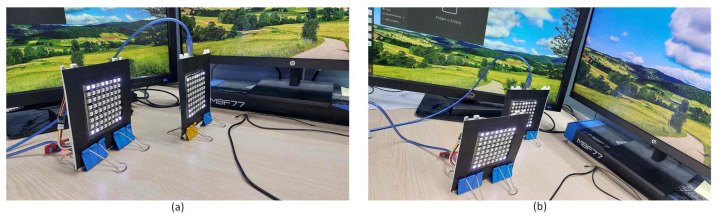
Two scenarios for the simulation. (**a**) The right-hand side transmitter placed ahead of the left-hand side transmitter, while (**b**) is the opposite of the first scenario.

**Figure 9 sensors-24-00702-f009:**
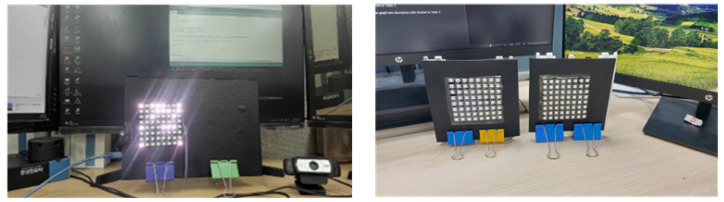
Sample images in the dataset used for YOLOv8 object detection model fine-tuning.

**Figure 10 sensors-24-00702-f010:**
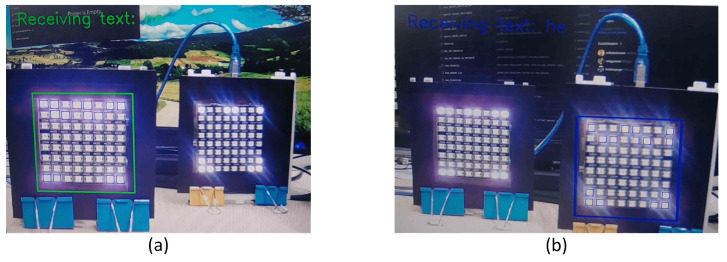
(**a**) A green square tracker on the left-hand side LED transmitter shows that transmission is in progress, while (**b**) a blue square tracker on the right-hand side LED transmitter shows that transmission is in progress.

**Figure 11 sensors-24-00702-f011:**
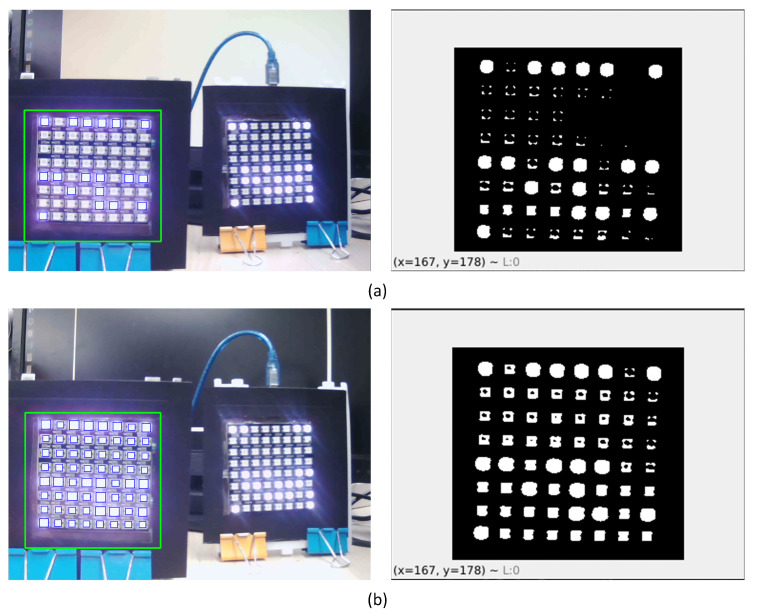
(**a**) shows better thresholding on a brighter background condition than (**b**). The latter figure has more noise due to lower brightness of the background.

**Table 1 sensors-24-00702-t001:** Hardware, software and hyperparameters used for the simulation.

Hardware	
CPU	Intel i7 12th gen
GPU	Nvidia RTX 3060Ti
RAM	16 GB
LED array	Neopixel 8 × 8
Camera	Foscam W41
**Software**	
Operating System	Ubuntu ver. 22.04
LED array controller	Arduino ver. 2.2.1
Framework	PyTorch ver. 2.1.1, ONNX ver. 1.16.0
**Hyperparameters**	
Learning rate	0.01
Batch size	1
Epoch	100

**Table 2 sensors-24-00702-t002:** Object detection algorithms performance comparison on LED dataset.

Model	mAP0.5:95	mAP0.5
YOLOv8	0.98251	0.83263
YOLOv5	0.97579	0.84668
Faster-RCNN (ResNet50)	0.98977	0.78301

**Table 3 sensors-24-00702-t003:** Comparison of BER performance at different communication distances.

Method	1 m	1.5 m	2 m	2.5 m	<5 m
OCC + DL [11]	$2\times 10^{-3}$	$2\times 10^{-3}$	$2\times 10^{-3}$	$2\times 10^{-3}$	$2\times 10^{-2}$
OCC + YOLOv5 [1]	$5\times 10^{-6}$	$5\times 10^{-6}$	$5\times 10^{-2}$	$5\times 10^{-2}$	$5\times 10^{-2}$
OCC + CNN [38]	$10^{-1}$	$10^{-1}$	$10^{-2}$	$10^{-3}$	$10^{-1}$
Proposed Model	$4.2\times 10^{-3}$	$4.15\times 10^{-3}$	$4.1\times 10^{-3}$	$4\times 10^{-3}$	$1.2\times 10^{-2}$

## Data Availability

Data are contained within the article.

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
