# Peer review of "Proximity-Based Optical Camera Communication with Multiple Transmitters Using Deep Learning"

_sensors, 2024, doi:10.3390/s24020702_

Round 1

Reviewer 1 Report

Comments and Suggestions for Authors

Dear Authors,

In the study, a 2D proximity-based method is proposed to determine the appropriate LED for communication. Thanks to this method, secure data transmission can be achieved in the OCC system. Although multiple transmitters may be present at the same time, data will only be available from the selected and designated transmitter. In the study, a method that prioritizes LED transmitters depending on the proximity of the transmitters to the receiver is introduced, a method for a switchable transmission system with multiple transmitters that uses 2D object size to measure the proximity of transmitters is proposed, and an approach to read LED array data without the need for camera parameter changes is described. The paper is an experimental study that contributes to the literature with the YOLOV8 image processing algorithm used. However, it would be appropriate to make the following corrections.

1. The accuracy rate obtained from the artificial intelligence algorithm used should be stated in the summary section.

2. In the introduction, reference should be made to new literature on Optical wireless communication. For example; "M. Açikgöz, et.al., "Optical Communication Infrastructure in New Generation Mobile Networks," FIBER AND INTEGRATED OPTICS, vol.42, no.2, pp.53-92, 2023. "

3. Why was the YOLOv8 algorithm used? Have you tried it with other YOLO algorithms? Performance comparison should be made using other image processing algorithms.

4. The photo quality and shooting method used in the experimental setup is not good. For example, it would be better if the finger was not visible in Figure 7. The clarity of Figures 7 and 9 can be improved.

5. The BER value of the system is below communication standards. This situation should be explained by the authors. In Figure 12, the axis names should be specified, and the graph should be drawn more professionally.

6. Have detection distance experiments been carried out? What is the safe operating distance of the system? (BER tests should be repeated according to distance)

7. How was the data set created for the algorithm? What is its size?

Best regards,

Comments on the Quality of English Language

Minor editing of English language required.

Author Response

Dear Reviewer,

We would like to thank you for the opportunity to resubmit a revised version. 
We have updated our manuscript according to your advice and highlighted it in yellow color. 
Please see the attachment below here.
Thank you very much.

Best regards,

Muhammad Rangga Aziz Nasution
Department of Electronics Engineering,
Kookmin University, Korea.

Reviewer 2 Report

Comments and Suggestions for Authors

The authors in the manuscript propose a scheme to improve optical camera communication. The method is proposed to fulfill the latter requirement using 2D object size to calculate the proximity of the objects through an AI object detection model. The author shows us the flow chart, hardware equipment, experimental steps and experimental results of the related technology in detail. Firstly, I would like to thank the authors for their contributions to this paper and secondly I would like to ask the following questions.

(1) Is the author's innovation related to the optical camera communication system structure or the object detection scheme or the algorithm improvement? I think it should be summed up more clearly at the end of the introduction.

(2) The first time the author mentions the YOLO target detector in the article should be briefly stated.

(3) In the analysis of bit error rate, the author does not involve the comparison of previous research experiments, so it is impossible to clearly judge whether the improvement scheme proposed by the author can make the system performance better than that of the predecessors.

(4) It is not comprehensive enough to analyze the system performance only by the bit error rate. For LED communication, it is more convincing to increase the analysis of relevant optical performance indicators.

(5) The communication distance mentioned in the article is 0.5-1m, can this distance be extended? If it is only applicable to a short distance of a few meters, I think it is of little significance to the actual optical camera communication improvement.

Comments on the Quality of English Language

 Minor editing of English language required

Author Response

(The authors gave the same response as above.)

Reviewer 3 Report

Comments and Suggestions for Authors

Thank you very much for the paper - interesting results and I think this would be interesting for the readers. What is important - theory is supported by the experiment. In fact I have not remarks or questions about the research you did.

Author Response

(The authors gave the same response as above.)

Round 2

Reviewer 1 Report

Comments and Suggestions for Authors

The authors made all suggestions and corrections.

Reviewer 2 Report

Comments and Suggestions for Authors

I think the revised manuscript can be published in this journal.